# The Role of Blue and Red Light in the Orchestration of Secondary Metabolites, Nutrient Transport and Plant Quality

**DOI:** 10.3390/plants12102026

**Published:** 2023-05-18

**Authors:** Alice Trivellini, Stefania Toscano, Daniela Romano, Antonio Ferrante

**Affiliations:** 1Department of Agriculture, Food and Environment, Università degli Studi di Catania, 95131 Catania, Italy; alice.trivellini@gmail.com; 2Department of Science Veterinary, Università degli Studi di Messina, 98168 Messina, Italy; stefania.toscano@unime.it; 3Department of Agricultural and Environmental Sciences—Production, Landscape, Agroenergy, Università degli Studi di Milano, 20133 Milan, Italy; antonio.ferrante@unimi.it

**Keywords:** light-emitting diodes, macronutrients, micronutrients, ornamental quality, nutritional status, horticulture

## Abstract

Light is a fundamental environmental parameter for plant growth and development because it provides an energy source for carbon fixation during photosynthesis and regulates many other physiological processes through its signaling. In indoor horticultural cultivation systems, sole-source light-emitting diodes (LEDs) have shown great potential for optimizing growth and producing high-quality products. Light is also a regulator of flowering, acting on phytochromes and inducing or inhibiting photoperiodic plants. Plants respond to light quality through several light receptors that can absorb light at different wavelengths. This review summarizes recent progress in our understanding of the role of blue and red light in the modulation of important plant quality traits, nutrient absorption and assimilation, as well as secondary metabolites, and includes the dynamic signaling networks that are orchestrated by blue and red wavelengths with a focus on transcriptional and metabolic reprogramming, plant productivity, and the nutritional quality of products. Moreover, it highlights future lines of research that should increase our knowledge to develop tailored light recipes to shape the plant characteristics and the nutritional and nutraceutical value of horticultural products.

## 1. Introduction

Light is a key environmental factor that affects many aspects of plant growth and development by regulating fundamental processes such as photosynthesis, primary and secondary metabolism, morphogenesis, and molecular and physiological responses [1]. Light represents the energy force driving the photosynthetic machinery to reduce carbon dioxide to carbohydrates. On the other hand, light also acts as a signal, and its dynamic perception and processing allows plants to synchronize and adjust their development to the ever-changing light environment. In particular, through the modulation of the light spectrum, it is possible to regulate light transduction mechanisms and manipulate specific plant characteristics, such as flowering induction, elongation, branching, secondary metabolites, nutrient status, seedling development, and cell growth [2]. Plants are naturally exposed to solar light, which includes the entire wavelength spectrum: ultraviolet (UV) (320–400 nm), blue (400–500 nm), green (500–600 nm), red (600–700 nm), and far-red (700–800 nm). Owning to their sessile nature, plants have evolved a sophisticated network of photoreceptors that sense different light wavelengths and, in a coordinated manner, activate upstream signaling cascades. The latter involves the massive reprogramming of a gene expression leading downstream to the activation of a number of light-specific complex physiological events and metabolic states [3,4]. Based on wavelengths they absorb, these multifunctional sensory proteins are classified into five classes: phototropins; cryptochromes and zeitlupes, which sense UV-A radiation or blue light; phytochromes, which absorb red and far-red light; and UV-B photoreceptor, UV RESISTANCE LOCUS 8 (UVR8) [5]. The light stimuli are perceived by photoreceptors prompting specific conformational changes and the activated sensors trigger a signal transduction to downstream modules/components of light signaling networks, which ultimately control light physiological and developmental processes in plants: i.e., PHYTOCHROME INTERACTING FACTORs (PIFs), CONSTITUTIVE PHOTOMORPHOGENIC 1 (COP1) and ELONGATED HYPOCOTYL 5 (HY5) [5]. PIFs are basic helix-loop-helix (bHLH) transcription factors that interact physically with phytochromes as regulatory hubs, controlling transcription machinery by binding to promoter regions (cis-element sites) of their target genes [6]. The spatiotemporal regulation of target gene expression (by induction or repression) orchestrates many plant developmental processes, including seed germination, flowering, photomorphogenesis, shade responses, and leaf senescence [7,8,9,10]. The other core light signaling regulatory network includes COP1 and HY5, where photoreceptors transduce the light signal to HY5 and modulate its activity through the transcriptional repression or activation of COP1, which acts as a negative regulator of HY5 function [11]. This interaction mediates cross-talk among multiple pathways that play critical roles in controlling seedling photomorphogenesis, shade avoidance, the circadian clock, root architecture, and flowering [12]. 

The responsiveness of plants to light signals and the understanding of how light signaling networks are connected to important plant productivity traits allow the strategical selection and activation of specific light-sensing pathways and building up specific light systems to control plant yield, quality, and production timing [5]. 

A closed plant production system with artificial lighting, where the growing environment is optimally controlled and all inputs supplied are fixed by plants with minimum emissions to the outside environment, is now attracting great attention in the horticultural industry and among researchers [13]. It offers several key advantages over conventional production systems: the growing artificial environment can be fabricated anywhere (no need for soil and solar radiation) and is not influenced by climate change factors; the productivity can be year-round, and the product can be pesticide-free with a longer shelf life. Moreover, these indoor hydroponic production systems have high resource use efficiency (water, agrochemicals, etc.), reducing the emission of pollutants as well as increasing the input use efficiency [13]. Finally, artificial lighting using light-emitting diode (LEDs) lamps creates a homogenous and efficient production environment at low costs, owning to their compact size, low lamp surface temperature, high light use efficiency, and broad light spectra. The possibility of selecting and controlling the intensity and wavelength of light allows the production of highly functional and cost-efficient plant products [14]. Specific light recipes have been reported to modulate the transcriptional machinery associated with diverse cellular functions, including those involved in micro- and macroelement uptake, transport, assimilation, and secondary metabolism, which are also supported by an increase in these nutrients and compounds in the plants [15,16,17,18,19]. Moreover, the ability to manage the light-growing environment through the selection of specific wavelengths offers the possibility to affect specific plant morphological traits which are reaching important productive advantages for the horticultural industry, such as early flowering, continuous production, predictable yield, and plant habitus (rooting and branching) [20]. 

This review provides an overview of the use of LEDs’ lighting technology in horticultural species, highlighting the effects of red and blue light wavelengths on decorative attributes, plant nutritional quality, and secondary metabolites (Figure 1).

## 2. Morphological Traits Regulation by LED Spectrum Quality

The horticultural industry includes plants from a vast range of species, such as fruits, vegetables, floriculture, and ornamental crops, and their marketability relies on species-specific plant traits which contribute to their superior quality. Selecting precise spectral compositions using LED technology offers the possibility to develop tailored light recipes for the manipulation of plant traits and thus obtain suitable plant characteristics [20]. Among the different light qualities, blue light and red light have been widely used for both research and commercial purposes because of their dynamic impact on morphogenesis, metabolism, and the capacity of photosynthesis as the absorption spectra of the photosynthetic pigments mainly focus on the blue (400–500 nm) and red (600–700 nm) light spectra [21]. Red light has an extensive impact on the photosynthetic process, promoting an increase in chlorophyll content but also restraining the mobilization of carbohydrates from source organs (i.e., leaves) [22,23]. With respect to physiological development, red light promotes cell division and extension, encouraging stem elongation [23,24]. Blue light enhances the chl a/b ratio and photosynthesis rate [25] and controls stomatal opening and the biosynthesis of secondary metabolites [26,27,28], while inhibiting cell division and extension of growth [23]. Horticulture plant quality takes advantage of LED lighting to shape plant architecture, induce flowering, prevent postharvest loss, and enhance flower and leaves pigmentation [29].

Flowering induction and transition can be efficiently triggered by the manipulation of the spectral composition of artificial lighting, emphasizing attractive features, thereby reducing production time, resulting in cost savings, obtaining predictable yield, and providing a competitive advantage in the long term. Plants sense changes in light environments and adjust their growth and development according to the photoperiod, light intensity, spectral composition, and light direction [30]. Plants are classified as long-day (LD) when flowering is promoted by night length less than a certain threshold, short-day (SD) when flowering is induced by long night, and day-neutral (ND) when flowering occurs independently of the length of daylight [25]. The use of specific wavelengths in combination with proper daylight duration can lead to the activation of transcriptional machinery which in turn drives flower transition [31]. The main photoreceptors involved in the perception and absorption of different spectral qualities are phytochrome, which preferentially absorbs in the red/far-red spectral regions, cryptochromes in the blue/UV-A wavelengths and phototropins (PHOT), ZTL/ FKF1/LKP2, and UVR8 mostly absorbs the UV-B wavelength [32]. At the molecular level, the perceived light signal for the conserved inductive photoperiod mechanism is transduced and triggers flower induction through the upregulation of FLOWERING LOCUS T (FT), also known as florigen and the repression of anti-florigenic FT (AFT)/TERMINAL FLOWER 1 (TFL1) in both LDs and SDs [33,34]. Regarding light quality, in photoperiodic plants, the regulatory role of blue, red, and far-red light is used to promote flowering and growth extension in long-day plants and short-day plants, respectively. The use of red/white/far-red light under a regime of daily light period exceeding their critical day length significantly enhanced the flowering transition in *Petunia hybrida* E.Vilm. and *Antirrhinum majus* L. plants [35]. In day-neutral *Cyclamen persicum* Mill., the combination of high light intensity and a spectrum comprising blue and red wavelengths boosted flower formation [36]. Moreover, a far-red-deficient light environment was used to prevent premature flowering in LDs (i.e., snapdragon, tussock bellflower, tickseed, and petunia) and promote their vegetative growth, assuring programmed flowering to predetermined market dates [37,38,39]. On the other hand, the interruption of the night with red light cycles is a widely used and cost-effective application for inhibiting the flowering of SDs based on the photochemical interconversion of phytochrome Pr to Pfr during night [40]. Similarly, blue wavelength at a high intensity has been reported to have an inhibitory action on short-day plants, whereas it showed a promoting flowering effect with an enhancement of the flowering index, visible flower buds, and opened flowers in long-day plants [41,42]. 

Artificial lighting helps shape plant architecture, representing a technological tool to stimulate branching, compactness, rooting, and leaf expansion by selecting blue and red wavelengths corresponding to the absorption spectra of photosynthetic pigments [21]. The production of young ornamental plants, including bedding plants, is realized during the winter season to guarantee spring–summer sales. During the production of these products in greenhouses, the photosynthetic daily light integral (DLI) is seasonally low. Thus, the use of supplemental light is a common practice to obtain more uniform, compact, and high-quality young horticultural plants with marketable characteristics. The application of blue radiation in the red background has been shown to delay growth and leaf expansion, ensuring a reliable tool for the control of height in various bedding plants [43,44,45]. Appropriate spectral qualities can positively influence the growth and survival of cuttings. Herbaceous perennial cuttings grown in a multilayer sole-source light propagation system equipped with red/blue light (R/B, 50:50) showed an increase in root biomass production and stem extension, which are valuable features for preventing damage during transport and transplantation [46]. Using blue in a strong red light background might have a positive control on transpiration, preventing the fast drying of cuttings, as blue light is responsible for stomatal opening response and effectively increases the number of trichomes that are linked to avoid the loss of water by transpiration [47,48]. In cut flower production, stem length is an important decorative trait for the marketability of a product, and at the physiological level, it is regulated by the availability of photosynthetically active radiation [49]. When the ratio of red/far-red (R/FR) decreases, a low phytochrome stationary state determines an increase in internode and petiole elongation, axillary bud outgrowth, and hyponasty [49]. Exposing the chrysanthemum plant to a blue and far-red light environment determined a higher internode length compared to the sole red light [50]. In lilium plants cultivated for cut flower industry, the exposure to the highest percentage of red in a blue light background (80 red:20 blue) has been shown to be effective in promoting the extensive elongation of the stems compared to the other light treatment [51]. In contrast, an increase in blue light percentage was associated with a more compact plant habitus, showing a better control of height and shoot length and an improvement in photosynthetic performance in several plant species [51,52,53]. Stem elongation of tomato plants grown in a controlled environment is affected by different red and blue spectral combinations and light intensity via the cryptochrome 2 photoreceptor [54]. Plant height decreased linearly with the increase in blue light proportion and the inhibition of stem elongation was higher at low light intensity, and this regulation may be mediated by the photoreceptor cryptochrome 2, as suggested by the correlation of plant height and the gene expression analysis [54].

## 3. Light Signaling Network of Nutrient Uptake and Utilization

Plants require 16 essential nutrients, among which, nitrogen (N), potassium (K), phosphorous (P), sulfur (S), magnesium (Mg), and calcium (Ca) are considered macroelements and are known to play a key role in the growth and development of plants [55]. In addition to the macronutrients iron (Fe), copper (Cu), zinc (Zn), manganese (Mn), boron (B), molybdenum (Mo), cobalt (Co), and nickel (Ni) are essential microelements required in small quantities by plants [55]. Plants have developed refined mechanisms to maximize nutrient absorption, redistribution, cellular compartmentalization, and assimilation to modulate cellular activity and development. Specific membrane transporters are involved in macro- and micronutrient uptake at the root–soil boundary [56]. Subsequently, absorption, endomembrane transporter systems operate to partitions nutrients between various intracellular compartments, whereas long-distance translocation among plant organs is achieved via bulk flow in the xylem and phloem vessels [56].

Several studies have highlighted the key role of light in the uptake and efficient use of essential elements for plant growth and development in response to fluctuating light environments [17]. Specifically, acquisition and use with growth and development are adjusted or altered by changes in light properties, such as quantity, duration, and quality. This modulation occurs after the perception of light stimuli and the subsequent activation of light signaling regulatory cascades by a series of sensory photoreceptors and downstream signaling components (COP1, PIFs, HY5) (Figure 2) [17,57]. Root nutrient absorption is orchestrated by the demand of shoots where photosynthesis occurs and relies on a root system with multiple signaling pathways to acquire the majority of nutrients [58]. These systemic root-to-shoot signals allows long-distance communication between distant plant organs, providing information on uptake capacity, transport, root growth toward soil resources, mineral remobilization, and/or partitioning at the whole-plant level [58].

Several long-distance components traveling between roots and shoots are involved in nutrient uptake and utilization and are thought to be directly linked to variations in spectral quality (i.e., blue and red light) [17]. The signal components that connect the signaling perception/transduction pathway of light with the subsequent optimization of nutrient absorption and utilization are phytohormones, Ca^2+^, sucrose, proteins, and microRNA (miRNA) [17,58]. Light signals continuously prompt plant growth and development remodeling and these dynamic adjustments are strictly regulated by phytohormones such as auxin, ethylene, cytokinins (CK), abscisic acid (ABA), gibberellins (GA), and brassinosteroids [59]. Morphogenetic modification modulated by the light environment involves the alteration of auxin homeostasis, since this hormone is a core regulator of growth and development [60]. Auxin is a mobile component that can be passively transported by carrier proteins over short or long distances in vascular tissues, synchronizing growth between the shoot and root through interaction with the light pathway [61]. The auxin efflux carrier PIN-FORMED (PIN) determines the direction of auxin flow out of the cell through its subcellular position on the plasma membrane [60]. In the dark, COP1 E3 ligase directly interacts with HY5 and targets it for proteasome-mediated degradation. On the other hand, light exposure deactivates COP1 and triggers HY5 expression, resulting in HY5 protein production [62]. HY5 acts as a signal integration point in the light and hormone signaling networks [63]. In shoots in the dark, COP1 has been reported to repress PIN expression by targeting HY5. Instead, the light signal prompts the transcriptional activation of PIN by COP1 deactivation in the root, allowing the transport of auxin from the shoot to the root and thus the elongation and/or root [64,65]. The auxin signaling network, which is activated by light and transmitted through the plant to the root tip, allows root elongation, which ultimately maximizes nutrient uptake [66]. The exogenous treatment of dark-grown *Arabidopsis* seedlings with IAA or tryptophan, a precursor of the auxin biosynthetic pathway (tryptophan), did not promote the formation of later roots [66]. However, an enhancement of the absorption radical area was reported, following treatment with an auxin synthetic analogous, NAA [66]. Red light has been demonstrated to encourage auxin flux from leaves to roots, possibly through the transcriptional regulation of PIN1, PIN3, and PIN4 genes [66]. On the other hand, when *Arabidopsis* plants were grown in a white light environment supplemented with far-red light, the lateral root density was lower. Adventitious root formation increases the surface area available for root absorption and is positively regulated by the blue light signaling pathways PHOT1 and PHOT2. In fact, both the number and density of adventitious roots and IAA concentrations were enhanced, suggesting a regulatory role of blue light in triggering adventitious root formation [66]. Moreover, evidence has shown that both the synthesis and transport of auxin are regulated by the absorption and compartmentalization of nutrients such as S and nitrate, suggesting a relationship between hormone and nutrient uptake [67,68]. Another class of phytohormones, brassinosteroids (BRs), has recently been linked to photomorphogenesis, skotomorphogenesis, and root development processes [69,70]. In the dark, skotomorphogenesis is achieved through the activation of the photomorphogenic repressor COP1–SPA complex, which in turn leads to the ubiquitination and degradation of downstream signaling components such as the photomorphogenesis-promoting transcription factors HY5, LONG HYPOCOTYL IN FAR RED (HFR1), GATA2, and B-BOX21 [70,71]. For example, the *Arabidopsis* G protein beta subunit (AGB1), a transcription factor involved in the regulation of light signaling and the brassinosteroid pathway, is known to act as a molecular switch to repress photomorphogenesis by limiting the transcriptional activity of BBX21 [72,73]. However, under blue light, CRY1 favors photomorphogenesis [72]. The photomorphogenesis is facilitated by photoactivated phyB under red light, which leads to PIF3 phosphorylation and degradation [73]. 

BRs play a key role in the balance of nutrient homeostasis and root growth as demonstrated by the exogenous application of BR biosynthesis inhibitors, which decreased P, K, Mg, Fe, and Mg concentrations [69]. The role of ABA–light interactions is evident in several physiological processes, such as seedling growth, shade avoidance, stem elongation, and leaf development [74]. ABA mediates light-regulated mineral uptake by modulating transporter proteins in root tissues. Exogenously applied ABA has been reported to inhibit the expression of the high-affinity plasma membrane Cu transporter, triggering the alteration of Cu [75] and Fe [76] homeostasis. 

Sugar signals and energy sensors have also been implicated in the dynamical regulation of nutrient uptake and utilization for metabolism, growth, and development [77]. A novel finding reported that an O-LINKED N-ACETYLGLUCOSAMINE (O-GlcNAc) TRANSFERASE (OGT) was actively involved in the suppression of DELLA protein, which in turn acts a transcriptional repressor by blocking the activity of bHLH transcription factors, such as PIFs affecting light-responsive gene expression [78]. Light is a critical factor in the activation of signaling integrators such as TOR, a conserved Ser/Thr protein kinase that orchestrates plant growth and development [77]. TOR activation, synergistically stimulated by sugars, energy status, nutrient accumulation (N, P, S), and light signaling (phytochromes and cryptochromes) promotes cell proliferation and elongation, such as soil exploration and the optimization of nutrient uptake by roots [17,77].

MicroRNAs (miRNAs) are a class of small (18–24 nt) single-stranded molecules that are evolutionarily conserved among many known species [79]. These endogenous non-coding particles mediate post-transcriptional gene expression by targeting the 3′ untranslated region (3′-UTR) of mRNA. This process acts as an efficient gene expression regulator, causes mRNA cleavage, and decreases protein translation [79]. Moreover, miRNAs act as mobile signals systemically throughout the plant and evidence has reported the role of these molecules in the regulation of nutrient transport and homeostasis [17]. Overexpression of miR399 modulates gene expression linked to nutrient acquisition and translocation, conferring enhanced tolerance to K- and P-deficient environments [80]. Other studies have shown that several miRNAs that regulate target genes involved in energy metabolism and secondary metabolic pathways are expressed in response to K deficiency, confirming the role of these molecules in nutrient homeostasis [81,82]. Another essential nutrient is S, whose uptake is strictly controlled of ATP sulfurylase enzyme. The gene encoding this enzyme is regulated by miR395 and its regulatory role in sulfate assimilation pathway was experimentally and computationally confirmed by Jones-Rhoades and Bartel [83] and Kawashima et al. [84]. In recent years, several studies have revealed a molecular basis integrating the transcriptional and post-transcriptional regulation of nutrient homeostasis and light signaling pathways [85]. The micronutrient Cu is an essential cofactor for several proteins and the most abundant plant Cu protein is plastocyanin (PC), which is involved in electron transfer from cytochrome b6f to photosystem I [85]. SQUAMOSA PROMOTER BINDING PROTEIN-LIKE7 (SPL7) is a key transcription factor that transcriptionally coordinates, including high-affinity Cu transporters [86] and miR397, miR398, miR408, and miR857 [85,87]. Combining RNA sequencing (RNA-seq) and chromatin immunoprecipitation sequencing (ChIP-seq) revealed that the SPL7 and HY5 transcription factors coregulate several genes linked to anthocyanin accumulation and photosynthesis [85] and act coordinately to regulate MIR408 at a transcriptional level. This finding suggests a role for SPL7/HY5/miR408 in mediating light–copper cross-talk. The major regulator in photomorphogenesis of the light signaling pathway, HY5, controls the expression of several miRNA genes by binding to the promoter region containing light-responsive elements [88]. Besides functioning as a structural component, cytosolic Ca^2+^ concentration plays a key role as a secondary messenger in numerous physiological processes where it modulates through its transient, sustained, or oscillatory enhancement downstream responses, including responses to abiotic and biotic stress, light stimuli and fertilization [17,89,90]. Changes in cytoplasmic Ca^2+^ contractions have been reported to occur in response to red, blue, and UVB light, which causes a fast transient increase in Ca [90,91,92].

## 4. Effect of Blue and Red Light on Nutrient Uptake and Utilization

Nutrient availability is fundamental for plant growth and development, and in recent years, several studies have generated a wealth of information on the dynamics of nutrient uptake, transport, and assimilation in response to changes in LED lighting in controlled environments that can be exploited to enhance crop productivity and quality. Specific wavelengths, such as red and blue light, are considered the main wavelengths needed for plant growth, which also influences plant resource absorption and allocation [17,93,94]. Nitrate occurs naturally in vegetables and particularly high levels are found in leafy vegetables such as lettuce and spinach [95]. Manipulation of the light spectrum can decrease the nitrate content to ensure compliance with the existing EU regulations and its limitation threshold [96]. Several studies have reported that red light or a combination of red and blue wavelengths generally decreases the nitrate concentration in leafy vegetables. In lettuce grown in a controlled environment, the short-term exposure of plants to red light at high intensity significantly lowers the concentration of nitrate [97], especially in expanded leaves and young leaves [98]. Similar results were obtained by the alternating use of red and blue light during the day [99], and by using a high proportion of red a blue light background [100]. In pak choi (*Brassica campestris* L.) grown under continuous illumination, the application of red or blue light significantly increased the activities of nitrate reductase (NR) and nitrite reductase (NiR) and their transcriptional expression showed a significant reduction in nitrate concentration, which might be partly attributable to the higher activity of NR [101]. The spectral variations of red and blue light have a substantial impact on nutrient uptake compared with the traditional lighting systems generally used. Hydroponically grown lettuce exposed to red/blue ratio (R/B, 3:1) showed an improved absorption of essential macronutrients such as N, P, K, and Mg from the nutrient solution, as a consequence of the increased plant biomass [102]. Similarly, indoor sweet basil plants grown under an R/B ratio of 3 showed an enhanced uptake of N, P, K, Ca, Mg, and Fe, suggesting a species-specific response to the light environment [103]. Overall, in both lettuce and basil, an R/B ratio of 3 guaranteed better performance in terms of biomass accumulation, nutrient use efficiency, and physiological functions compared to the other light treatments [102,103]. 

In celery, higher ratios of red to blue light (R/B; 4:1 and 7:1) have been shown to result in the highest Zn and Se concentrations, respectively [59]. The equal ratio of blue and red light in mulberry seedlings determined a significant enhancement of Mn and Cu concentrations, while the content of Zn reached the maximum value when the percentages of red/blue light were 70 and 30%, respectively [59]. However, exposure to a higher blue and red light ratio (5:1), despite the increased concentration of nutrients such as N, Mg, Zn, and Cu, negatively affects biomass accumulation and leaf growth in lettuce plants [104].

The spectral combination of far-red, red, and blue in hydroponically grown lettuce plants led to the highest growth and mineral uptake (K, Ca, and Mg) compared to control plants grown under conventional white light HPS lamps [105]. The combination of red and blue wavelengths has been shown to enhance K uptake in plant roots through the modulation of K transporter genes, whose expression levels are upregulated as a result [106].

Increasing the proportion of blue light relative to red light resulted in a higher content of all mineral nutrients in mustard and kale microgreens [107]. In sprouting broccoli (*Brassica oleacea* L. var. *italica* Plenck), the use of higher ratios of red/blue light (R/B; 1:4) determined the highest levels of Ca, Mg, P, S, B, Cu, Fe, Mn, Mo, and Zn [108]. Moreover, the nutrient concentrations of N, P, Mg, Fe, and Zn in einkorn seedlings significantly increased under higher blue/red ratios (B/R; B50:R50 and B75:R50) compared to white, monochromatic red, and blue lights [109]. Similar results were reported for lettuce and basil plants where a higher percentage of blue light resulted in increases in S, Mg, and B [110], and in Ca [111], respectively. The positive effects of higher blue light on nutrient uptake seem to be species-specific, dependent on a particular element as well as on different blue/red ratios. For example, both mustard and kale microgreens had high concentrations of Ca, P, B, and Mg under high blue light ratios (B50:R50, B75:R25, and B100:R0). High S content was found only in mustard, a high content of Cu, Zn, and K was found only in kale, and a high content of all nutrients was found in kale under the B75:R25 ratio [108]. The physiological basis of blue light properties relies on the control of the phototropin receptor exerted by blue light, which promotes an ion channel opening, encouraging ion transport [17,112,113]. In addition, monochromatic red has been effective in increasing the accumulation of N and Mg in lettuce leaves [111]. Similarly, in okra, the highest Ca, P, and Mn values were obtained under red light alone [114]. When red wavelength was used as a supplemental intra-canopy LED lighting source in the greenhouse, K, Mg, and Ca content in whole tomato fruit increased by 30, 74, and 40%, respectively, compared to HPS lamps [115], representing an interesting tool for improving fruit quality during off-season production. Red light has been reported to have a positive impact on K absorption in cucumber and spinach, whereas the content of K was found to be lower under blue light [17,116]. On the other hand, in Chinese chive and garlic, blue light enhanced K uptake [17]. Table 1 summarizes the effects of blue and red lights on nutrient uptake. Supplemental red and blue (LED) light in hydroponically grown tomatoes was evaluated for K uptake and transport as well as fruit coloring [106], showing that K accumulation was enhanced by the use of a red light source in combination with blue (R/B, 75%:25%) through the transcriptional regulation of K transporter genes [106]. These data suggest that K transport in tomato fruits may be mediated by red, far-red, and blue light signaling.

## 5. Effect of Light Spectrum on Plant Bioactive Compounds

Light quality is not only involved in plant photomorphogenesis and mineral uptake, but has also been shown to induce the synthesis of secondary metabolites and antioxidants via the photosensory network driven by photoreceptor pathways, allowing the production of nutraceutical and nutritionally enriched plant products [126,127]. Several studies have investigated the regulatory role of light quality treatments, red (R) and blue (B) wavelengths and their ratios, on secondary metabolism to modulate the concentration of functional metabolites in many horticultural crops [20]. The use of blue and/or red wavelengths is an essential environmental factor for anthocyanin biosynthesis and accumulation. In strawberries, blue and red light treatments resulted in significantly higher levels of total anthocyanins with the highest and fastest stimulation obtained under blue light exposure [117]. Red light in addition to the enhancement of anthocyanin concentration promotes the synthesis of proanthocyanidins through the upregulation of leucoanthocyanidin reductase (LAR) and anthocyanidin reductase (ANR) genes [117].

In lettuce, the use of a single-spectral blue or red LEDS, or a combination of both has been shown to increase biomass accumulation and anthocyanin concentrations in these leafy vegetables [128,129]. The application of red light as a sole-source lighting system in coriander plants leads to a significant decrease in antioxidant properties compared to the three ratios of red to blue applied [118]. In general, various bioactive compounds respond differently to light treatment and their biosynthesis is expected to be promoted in a species/cultivar-specific manner. For example, the effect of light spectra on vitamin C has been reported to differ between the two lettuce cultivars [119]. The vitamin C content increased significantly with red/blue (1:1) treatment in red lettuce, whereas the same light exposure showed a negative impact in green lettuce [119]. Similarly, in several microgreen species, the improvement of phytochemical content appears to be strongly dependent on the species and specific spectral wavelengths applied, boosting the functional and nutraceutical quality of these products [120]. The biosynthesis and accumulation of phenolic compounds and anthocyanins were strongly induced under a higher percentage of red light in Brassicaceae microgreens. The use of an increased red light proportion over a blue light background enhanced phenol accumulation and antioxidant capacity displayed by ferric-reducing antioxidant power (FRAP) in five of seven microgreens [130]. Overall, secondary metabolism can be modulated by different light spectra to produce phytochemically enriched products. Moreover, the use of an LED lighting environment is exploited to prove the quality of products after their harvest because the functional and biological roles of secondary metabolites vary. In lettuce, red light has been shown to promote moisture retention, avoiding quick water evaporation after harvest, thus preserving the market and quality acceptability [131,132]. Pepper-harvested fruit illuminated with higher blue light fractions resulted in a higher anthocyanin synthesis and showed a delay in the ripening process supported by the downregulation of senescence-related genes [121]. In cut carnation flowers, the application of a sole-source blue light spectrum under cold storage has been reported to prolong vase life, increase antioxidant capacity, and markedly delay senescence processes by reducing the expression of ACS1 and ACO1 genes involved in ethylene biosynthesis [122,123]. Furthermore, the manipulation of LED spectra during cultivation might be a useful tool to boost secondary metabolites and antioxidant capacity and obtain enriched by-products. *Crocus sativus* L. plants grown under a combination of red and blue light showed a significant enhancement of antioxidant compounds in tepals (flavonoids, flavonols, flavonol glycosides, and antioxidant capacity) compared to plants grown under natural light in a greenhouse [124]. *C. sativus* tepals are generally discarded during saffron production but they could be valorized and considered as a novel functional ingredient for food and cosmeceutical industries using LED light treatments. Supplemental red and blue (LED) light in hydroponically grown tomatoes was evaluated for nutraceutical quality and fruit coloring [106]. Blue or red combined with blue, led through the regulation of K transporter genes, has been reported to enhance the concentration of phytoene, β-carotene, α-carotene, and γ-carotene content and accelerate fruit coloring during fruit ripening [106]. In addition to stimulating the antioxidant defense system, the modulation of light spectra has been reported to regulate heavy metals tolerance. In cucumber, blue and red light antagonistically balanced Cd tolerance, through the modulation of photosynthesis, antioxidant defense systems, and Cd uptake [133]. The mitigation of Cd-induced oxidative damage was achieved by the application of blue light, which has been reported to improve the activities of antioxidant enzymes, reduce the content of radical oxygen species, downregulate the expression levels of Cd uptake and transport genes, and reduce Cd content in plant tissue [133]. Table 1 summarizes the effects of blue and red light on secondary metabolite concentrations.

## 6. Conclusions

Light is an essential environmental factor that coordinates plant growth, development, physiological and metabolic processes from flowering, architectural traits, nutrient uptake and assimilation, and secondary metabolites. Red and blue light wavelengths have a dynamic impact on morphogenesis, metabolism, and a capacity of photosynthesis because the absorption spectra of the photosynthetic pigments mainly focus on the blue (400–500 nm) and red (600–700 nm) light spectra. LED technology has the potential to develop tailored light recipes to shape plant quality traits, and thus obtain suitable plant characteristic through the modulation of flowering induction, branching, compactness, rooting, and leaf expansion. Blue and red light are actively involved in nutrient absorption and assimilation by a complex signal transduction pathway linked to light perception, and the synthesis and accumulation of bioactive compounds can be effectively driven by the manipulation of spectral light components. This review highlights the morphological traits, the accumulation of important plant metabolites, and the dynamic nutrient uptake signaling networks, including plant hormones that are orchestrated by blue and red wavelengths, with a focus on transcriptional and metabolic reprogramming, plant productivity, and nutritional quality of products. Nutrient uptake is directly correlated with photosynthesis and, hence, light intensity. Nitrate accumulation is regulated by leaf light exposure and nitrate reductase activity. The accumulation of bioactive compounds can help plants to counteract abiotic stresses and can be appreciated by consumers for their positive effects on health. The effect of light on secondary metabolites increases the antioxidant potential of the products through an increase in phenols. The use of LED technology properly balanced in red/blue proportions can effectively improve the overall plant quality in terms of aesthetic, nutraceutical, and nutritional properties.

## Figures and Tables

**Figure 1 plants-12-02026-f001:**
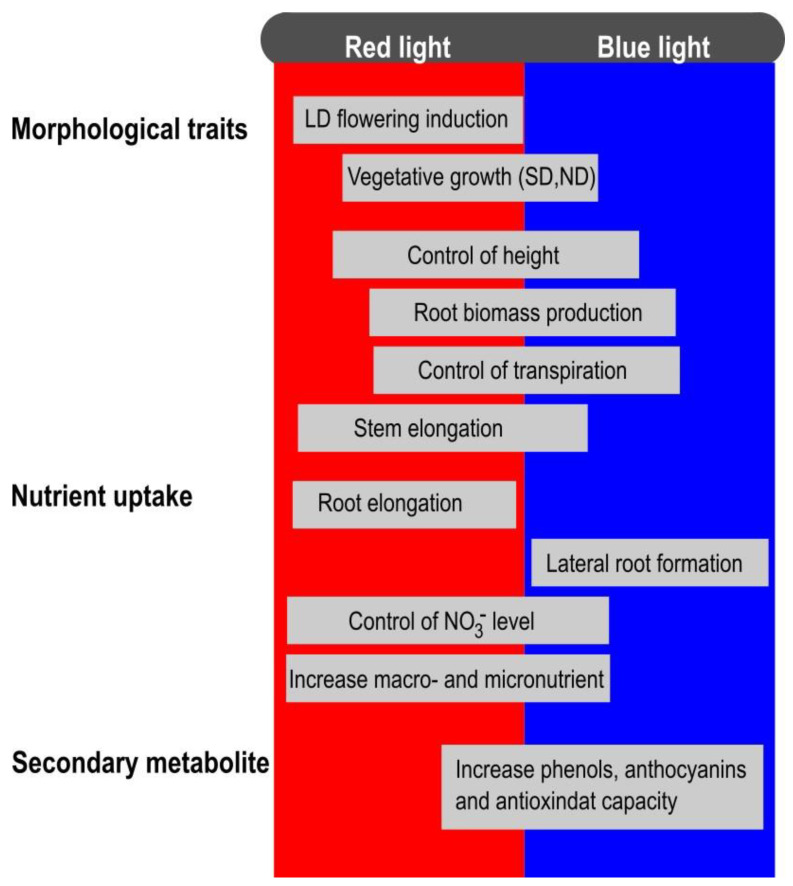
Overview of the effects of blue and red light on morphological traits, nutrient uptake, and secondary metabolite.

**Figure 2 plants-12-02026-f002:**
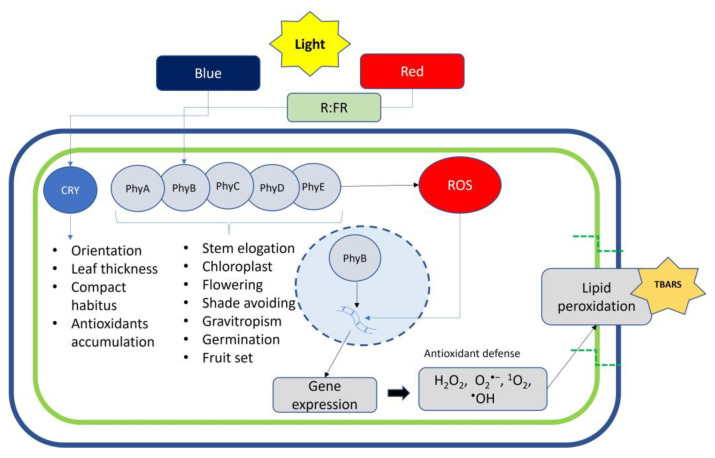
Perception of red and blue wavelengths and downstream signaling transduction network regulating plant physiology.

**Table 1 plants-12-02026-t001:** Effects of red and blue light wavelength on macro- and micronutrient content level and secondary metabolites accumulation in horticultural crops grown in a controlled environment.

Light Wavelength	Crop Species	Content Level		References
Red light	*Lactuca sativa* L.	**Nutrients**	NO_3_^−^ **↓**	[97]
R/B not reported	*Lactuca sativa* L.		NO_3_^−^ **↓**	[100]
Red light	*Brassica campestris* L.		NO_3_^−^ **↓**	[101]
Blu light	*Brassica campestris* L.		NO_3_^−^ **↓**	[101]
R/B = 3:1	*Lactuca sativa* L.		N P K Mg **↑**	[101]
R/B = 3:1	*Ocimum basilicum* L.		N P K Ca Mg Fe **↑**	[103]
R/B = 4:1	*Apium graveolens* L.		Zn **↑**	[17]
R/B = 7:1	*Apium graveolens* L.		Se **↑**	[17]
FR/R/B = not reported	*Lactuca sativa* L.		K Ca Mg **↑**	[103]
R/B = 1:3	*Brassica juncea* (L.) Czern.		P K Ca Mg S Mn **↑** Fe Zn Cu B	[107]
R/B = 1:3	*Brassica napus* L.		P K Ca Mg S Mn **↑** Fe Zn Cu B	[107]
R/B = 4:1	*Brassica oleracea* L.		Ca Mg P S B Cu **↑** Fe Mn Mo Zn	[107]
R/B = 1:1.5; 1:3	*Triticum monococcum* L.		N P Mg Fe Zn **↑**	[109]
Blue light	*Fragaria × ananassa* Duchesne ex Decne. and Naudin	**Secondary metabolites and antioxidants**	Anthocyanins **↑**	[117]
Red light	*Fragaria × ananassa* Du-chesne ex Decne. and Naudin		Anthocyanins, **↑** proanthocyanidins	[117]
Red light	*Coriandrum sativum* L.		Antioxidant **↑** capacity	[118]
R/B = 1:1	*Lactuca sativa* L. green		Ascorbic acid **↓**	[119]
R/B = 1:1	*Lactuca sativa* L. red		Ascorbic acid **↑**	[119]
R/B = 9:1	*Raphanus raphanistrum* L.		Phenols **↑**	[120]
R/B = 9:1	*Ocimum basilicum* L.		Phenols, **↑** antioxidant capacity	[120]
R/B = 9:1	*Amaranthus tricolor* L.		Phenols, **↑** antioxidant capacity	[120]
R/B = 9:1	*Allium schoenoprasum* L.		Phenols, **↑** antioxidant capacity	[120]
R/B = 9:1	*Borago officinalis* L.		Phenols, **↑** antioxidant capacity	[120]
R/B = 9:1	*Pisum sativum* L.		Phenols, **↑** antioxidant capacity	[120]
R/B = 1:3	*Capsicum annuum* L.		Anthocyanins **↑**	[121]
Blue light	*Dianthus caryophyllus* L.		Antioxidant **↑** capacity	[122,123]
R/B = 1.6:1	*Crocus sativus* L.		Flavonoids, **↑** flavonols, antioxidant capacity	[124]
Red light	*Amaranthus tricolor* L.		Phenols **↓**	[125]
Blue light	*Amaranthus tricolor* L.		Phenols, ascorbic acid, antioxidant capacity **↑**	[125]
Red light	*Brassica rapa* L. subsp. *oleifera* (DC.) Metzg		Ascorbic acid **↑** antioxidant capacity **↓**	[125]
Blue light	*Brassica rapa* L. subsp. *oleifera* (DC.) Metzg		Phenols, ascorbic acid, antioxidant capacity **↑**	[125]

↓ = decrease; ↑ = increase.

## Data Availability

Not applicable.

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
