# Peer review of "The Role of Blue and Red Light in the Orchestration of Secondary Metabolites, Nutrient Transport and Plant Quality"

_plants, 2023, doi:10.3390/plants12102026_

Round 1
Reviewer 1 Report
The topic of plant quality & accumulation of secondary metabolites under treatment with different LED light is essentially a demanding field of plant nutrition and secondary metabolism and this research, by no doubt, which will attract a wide audience in the journal. The authors did an amazing effort for collecting huge quantity of articles and presented an ambitious attempt to coordinate this review. Although there are some similar studies already published, data yielded by this comprehensive review manuscript has proper length and robustness. I believe that this article may become more informative and authoritative for readers after a major revision in terms of languae, recent information and additional articles as the example one given below:
Oana L., Carmen M., Behnaz R., and Calina P.C., LED Technology Applied to Plant Development for Promoting the Accumulation of Bioactive Compounds: A Review. Plants 2023, 12, 1075. https://doi.org/10.3390/plants12051075
And, more articles are available in Google Scholar search.
I have another concern in the Fig 1, if it is directly taken from a published source, for example reference [17] you should submit permission from previous authors/publisher in order to avoid future consequences. You should mention it here (as a reply to reviewer comment), in written, that you didnt take directly or you have permission.
Author Response
Reviewer 1
Dear reviewer,
The authors would like to thank you for your comments. The manuscript has been accordingly revised. Corrections and suggestions have been implemented in the current version of the manuscript. All the modifications are highlighted in yellow in the manuscript. We hereby provide a point-by-point answer.
The authors
The topic of plant quality & accumulation of secondary metabolites under treatment with different LED light is essentially a demanding field of plant nutrition and secondary metabolism and this research, by no doubt, which will attract a wide audience in the journal. The authors did an amazing effort for collecting huge quantity of articles and presented an ambitious attempt to coordinate this review. Although there are some similar studies already published, data yielded by this comprehensive review manuscript has proper length and robustness. I believe that this article may become more informative and authoritative for readers after a major revision in terms of languae, recent information and additional articles as the example one given below:
Oana L., Carmen M., Behnaz R., and Calina P.C., LED Technology Applied to Plant Development for Promoting the Accumulation of Bioactive Compounds: A Review. Plants 2023, 12, 1075. https://doi.org/10.3390/plants12051075
And, more articles are available in Google Scholar search.
Author Answer (A.A.): thank you very much for the positive comments and suggestions. As suggested by this reviewer and previously ones we included and discussed more recent research in this manuscript. The suggested paper was included.
I have another concern in the Fig 1, if it is directly taken from a published source, for example reference [17] you should submit permission from previous authors/publisher in order to avoid future consequences. You should mention it here (as a reply to reviewer comment), in written, that you didnt take directly or you have permission.
A.A.: We cited several times the well written Xu et al paper. The Figure 1 of this manuscript was not taken directly from Xu et al paper, we drew it following the paragraph content. Our figure is more detailed even though some similarities exist.
Reviewer 2 Report
The manuscript is well-prepared and is worth publication.
Author Response
Reviewer 2
Dear reviewer,
The authors would like to thank you for your comments. The manuscript has been accordingly revised. Corrections and suggestions have been implemented in the current version of the manuscript. All the modifications are highlighted in yellow in the manuscript. We hereby provide a point-by-point answer.
The authors
Comments and Suggestions for Authors
The manuscript is well-prepared and is worth publication.
Author Answer (A.A.): Thank you very much for your positive evaluation.
Reviewer 3 Report
The review by Trivellini and colleagues deals with the effect of blue and red LED light on aesthetic, morphological, nutritional properties, etc. of horticulture plants and how changes in the LED composition or ratio can be used to find optimal properties or advantages from a commercial, nutritional or nutraceutical viewpoint. The review is of interest for plant biologists and photobiologists and it should be considered for publication after undertaking the following points:
Lines 2 and 3
The title should be improved. Plant quality, secondary metabolites, nutrient transport, etc. are responses/effects mediated or orchestrated by blue and red light. So, the authors should use the past tense “mediated or orchestrated” to be more precise. Alternatively, the authors could start the title writing “On the role of blue and red LED light in …”
Line 19
Please, spell out “HPS”
Lines 23-28
The final part of the abstract should be more focused on the specific impact blue and red LED light on plant quality, secondary metabolites, nutrient transport, etc instead of giving simply general statements of the light effect without any specification. The readers are expecting more information related to the title of the review.
Please, use throughout the text radiation when referring to UV radiation and light when referring to the visible part of the electromagnetic radiation.
Lines 115-117
The authors should provide references in which LED lighting was specifically used to sustain their statement.
Line 185
Write the symbol for calcium as it has been done for the rest of nutrients
Lines 231 and 232
References should follow the journal format style according to the instructions for authors
Lines 183-299
Section 3 is mainly focused on light-mediated signalling networks with no emphasis in responses/effects mediated by red or blue LED light or a combination of both. Only in lines 298-299 different changes in cytoplasmic Ca concentration is briefly described in response to different types of radiations. So, the authors should make the effort to describe differences in the signalling network depending on the use of blue or red LED light and reduce the length of this section if it is not really focused on red and blue LED light signalling networks.
Line 378 and 386
Do not introduce abbreviations that are not used throughout the text or are only used once.
Line 389-391
Should not the arrow in Table 1 for Coriandrum sativum L. be pointing down based on the sentence written in these lines?
English style should be revised, particularly the order of the subject of the sentences and other complements.
Author Response
Reviewer 3
Dear reviewer,
The authors would like to thank you for your comments. The manuscript has been accordingly revised. Corrections and suggestions have been implemented in the current version of the manuscript. All the modifications are highlighted in yellow in the manuscript. We hereby provide a point-by-point answer.
The authors
Comments and Suggestions for Authors
The review by Trivellini and colleagues deals with the effect of blue and red LED light on aesthetic, morphological, nutritional properties, etc. of horticulture plants and how changes in the LED composition or ratio can be used to find optimal properties or advantages from a commercial, nutritional or nutraceutical viewpoint. The review is of interest for plant biologists and photobiologists and it should be considered for publication after undertaking the following points:
Author Answer (A.A.): thank you very much for the positive comments and suggestions
Lines 2 and 3
The title should be improved. Plant quality, secondary metabolites, nutrient transport, etc. are responses/effects mediated or orchestrated by blue and red light. So, the authors should use the past tense “mediated or orchestrated” to be more precise. Alternatively, the authors could start the title writing “On the role of blue and red LED light in …”
A.A.: As suggested by the referee the title was modified
Line 19
Please, spell out “HPS”
A.A.: We delete this sentence since was not relevant
Lines 23-28
The final part of the abstract should be more focused on the specific impact blue and red LED light on plant quality, secondary metabolites, nutrient transport, etc instead of giving simply general statements of the light effect without any specification. The readers are expecting more information related to the title of the review.
A.A.: following reviewer suggest the abstract was modified.
Please, use throughout the text radiation when referring to UV radiation and light when referring to the visible part of the electromagnetic radiation.
A.A.: As suggested by the reviewer we followed his/her suggestion
Lines 115-117
The authors should provide references in which LED lighting was specifically used to sustain their statement.
A.A.: As suggested by the reviewer we added the refence
Line 185
Write the symbol for calcium as it has been done for the rest of nutrients
A.A.: We modified it following reviewer suggestion
Lines 231 and 232
References should follow the journal format style according to the instructions for authors
A.A.: According to the referee we modified them.
Lines 183-299
Section 3 is mainly focused on light-mediated signalling networks with no emphasis in responses/effects mediated by red or blue LED light or a combination of both. Only in lines 298-299 different changes in cytoplasmic Ca concentration is briefly described in response to different types of radiations. So, the authors should make the effort to describe differences in the signalling network depending on the use of blue or red LED light and reduce the length of this section if it is not really focused on red and blue LED light signalling networks.
A.A.: following reviewer comment we incorporated the interaction/effect between blue and red on signaling network
Line 378 and 386
Do not introduce abbreviations that are not used throughout the text or are only used once.
A.A.: following referee suggestion we eliminated the abbreviations.
Line 389-391
Should not the arrow in Table 1 for Coriandrum sativum L. be pointing down based on the sentence written in these lines?
A.A.: As suggested by the reviewer we corrected it and pointed the arrow down.
Comments on the Quality of English Language
English style should be revised, particularly the order of the subject of the sentences and other complements.
A.A.: The English editing has been performed using PaperPal Pro.
Reviewer 4 Report
The topic of this review paper is well-selected: due to the spread of LED technology, investigation of different spectra on various plant physiological processes is a fashionable work nowadays. Therefore, present review paper may be interesting for wide range of readers. Basically, the paper is well-written; however, there are a few things, which must be modified.
- This is a kind of hot topic, still, only less than 10% of the references are from the last few years (2021-23). Much more recent papers with novel findings should be cited. The authors should focus on the novelties, newly discovered mechanisms and relationships, instead of general knowledge.
- The only figure they showed is only good for a graphical abstract, but does not reflect the recent findings on the mechanisms. This must be replaced with a more detailed version, focusing on the recent findings. Furthermore, since the paper basically has 3 main subtopics, including light and morphology, light and nutrient uptake, and light and secondary metabolism (bioactive compounds); all of these should be presented in the figure (or even better in separate figures).
- A few important aspects are missing, which should also be mentioned in a subchapter, or at least in separate paragraphs. These are: i. the authors only focused on the spectral composition. However, the effects are also depend on the intensity, so it should also be discussed. This is an important topic, because from practical point of view, it must be known how the intensity can be replaced with the spectral changes (see for example, Utasi et al., Scientia Hort. 2023; Fan et al., 2013, Sci Hort. etc.); ii., although this paper focuses on controlled conditions, stress aspects should also be mentioned. They are especially related to metabolic processes.
- The Conclusion must be completed with novel ideas and future aspects, recommendation. Its present form is very general without any forward-looking parts.
Author Response
Reviewer 4
Dear reviewer,
The authors would like to thank you for your comments. The manuscript has been accordingly revised. Corrections and suggestions have been implemented in the current version of the manuscript. All the modifications are highlighted in yellow in the manuscript. We hereby provide a point-by-point answer.
The authors
Comments and Suggestions for Authors
The topic of this review paper is well-selected: due to the spread of LED technology, investigation of different spectra on various plant physiological processes is a fashionable work nowadays. Therefore, present review paper may be interesting for wide range of readers. Basically, the paper is well-written; however, there are a few things, which must be modified.
This is a kind of hot topic, still, only less than 10% of the references are from the last few years (2021-23). Much more recent papers with novel findings should be cited. The authors should focus on the novelties, newly discovered mechanisms and relationships, instead of general knowledge.
Author Answer (A.A.): thank you very much for the positive comments and suggestions. As suggested by the reviewer we did our best to add novel findings
- The only figure they showed is only good for a graphical abstract, but does not reflect the recent findings on the mechanisms. This must be replaced with a more detailed version, focusing on the recent findings. Furthermore, since the paper basically has 3 main subtopics, including light and morphology, light and nutrient uptake, and light and secondary metabolism (bioactive compounds); all of these should be presented in the figure (or even better in separate figures).
A.A.: Due the complexity of signal transduction pathway we prefer simplify the perception of red and blue wavelength and downstream signaling transduction network regulating macro- and micronutrient absorption following the paragraph essential concept. Regarding the other three main subtopic they were summed up in table 1.
- A few important aspects are missing, which should also be mentioned in a subchapter, or at least in separate paragraphs. These are: i. the authors only focused on the spectral composition. However, the effects are also depend on the intensity, so it should also be discussed. This is an important topic, because from practical point of view, it must be known how the intensity can be replaced with the spectral changes (see for example, Utasi et al., Scientia Hort. 2023; Fan et al., 2013, Sci Hort. etc.); ii., although this paper focuses on controlled conditions, stress aspects should also be mentioned. They are especially related to metabolic processes.
A.A.: As suggested by the reviewer the effect of light intensity and stress topic were also incorporated in the manuscript.
- The Conclusion must be completed with novel ideas and future aspects, recommendation. Its present form is very general without any forward-looking parts.
A.A.: As suggested by the reviewer the conclusion was modified
Round 2
Reviewer 3 Report
The authors adequately address the main comments raised by the reviewer and now the review is recommended for publication.
Very minor typos are dectected that should be removed after editorial revision.
Author Response
Response to Reviewer 3 Comments
Comments and Suggestions for Authors
The authors adequately address the main comments raised by the reviewer and now the review is recommended for publication.
Author Answer (A.A.): Thank you very much for your positive evaluation.
Comments on the Quality of English Language
Very minor typos are dectected that should be removed after editorial revision.
A.A.: We have tried our best to control the linguistic aspects
Reviewer 4 Report
The MS has been improved; however, I am still not satisfied with the Figure. It only shows that although through different photoreceptors, blue and red light have the same effects. This is not entirely true, so this figure must be modified, and the similarities and differences must also be pointed out. If the authors are able to present a more complex figure demonstrating the recent findings and most important relationships, I will be ready to recommend its acceptance. The table is also informative in its present form; however, it has a descriptive nature.
Author Response
Response to Reviewer 4 Comments
Comments and Suggestions for Authors
The MS has been improved; however, I am still not satisfied with the Figure. It only shows that although through different photoreceptors, blue and red light have the same effects. This is not entirely true, so this figure must be modified, and the similarities and differences must also be pointed out. If the authors are able to present a more complex figure demonstrating the recent findings and most important relationships, I will be ready to recommend its acceptance. The table is also informative in its present form; however, it has a descriptive nature.
Author Answer (A.A.): Thanks for the comments and suggestions. Based on your comments, the figure 1 has been revised and modified by adding the relationships from a morphological, nutrient absorption and secondary metabolite point of view. We hope that the new figure meets the required requirements. A second figure has been added (Figure 2): Perception of red and blue wavelengths and downstream signaling transduction network regulating plant physiology. This figure has been added to highlight the modulation that occurs after the perception of light stimuli and the subsequent activation of light signaling regulatory cascades by a series of sensory photoreceptors and downstream signaling components (COP1, PIFs, HY5).
All new variations are highlighted in green in the manuscript.